# Hypertriglyceridemia and Other Risk Factors of Chronic Kidney Disease in Type 2 Diabetes: A Hospital-Based Clinic Population in Greece

**DOI:** 10.3390/jcm11113224

**Published:** 2022-06-06

**Authors:** Ilias N. Migdalis, Ioannis M. Ioannidis, Nikolaos Papanas, Athanasios E. Raptis, Alexios E. Sotiropoulos, George D. Dimitriadis

**Affiliations:** 1Second Medical Department and Diabetes Centre, NIMTS Hospital, 115 21 Athens, Greece; 2First Medical Department and Diabetes Centre, Hospital of Nea Ionia Konstantopoulio-Patision, 142 33 Athens, Greece; driioann@gmail.com; 3Second Department of Internal Medicine and Diabetes Centre, University Hospital of Alexandroupolis, Democritus University of Thrace, 681 00 Alexandroupolis, Greece; papanasnikos@yahoo.gr; 4Second Department of Internal Medicine, Research Institute and Diabetes Centre, Attikon University Hospital, Medical School, National and Kapodistrian University of Athens, 124 62 Athens, Greece; atraptis@med.uoa.gr (A.E.R.); gdimitr@med.uoa.gr (G.D.D.); 5Third Medical Department and Diabetes Centre, Hospital of Nikaia, 184 54 Piraeus, Greece; alexiosotiropoulos@gmail.com

**Keywords:** type 2 diabetes mellitus, blood pressure, triglycerides, diabetic chronic kidney disease

## Abstract

Aims/Introduction: Several reports indicate an increasing prevalence of chronic kidney disease (CKD) in type 2 diabetes mellitus (T2DM). Hyperglycemia and hypertension are the main risk factors for CKD development and progression. However, despite the achievement of recommended targets for blood glucose and blood pressure (BP), the residual risk of diabetic chronic kidney disease (DCKD) remains relatively high. The aim of this study is to examine dyslipidemia and other major risk factors to provide support for the prevention and treatment of DCKD. Materials and Methods: Participants are from the Redit-2-Diag study that examines 1759 subjects within a period of 6 months. DCKD severity is staged according to KDIGO criteria. Results: An increase in hemoglobin A_1c_ (1 unit) and systolic blood pressure (1 mm Hg) increases the probability of being classified into a higher CKD stage by 14% and 26%, respectively. Moreover, an increase of triglycerides by 88.5 mg/dL increases the risk of classification to a worse CKD stage by 24%. Conclusions: Elevated triglycerides, systolic blood pressure, and poor glycemic control increase the risk of CKD in T2DM and should be addressed in the treatment strategies.

## 1. Introduction

Type 2 diabetes mellitus (T2DM) and hypertension are the leading risk factors for the development and progression of chronic kidney disease (CKD) [1]. Previous studies have estimated a 27.9–63.9% diabetic chronic kidney disease (DCKD) prevalence in T2DM [2]. In Greece, we have recently reported a 45% overall prevalence of DCKD (mild, moderate, and severe) in a hospital-based T2DM population followed in diabetes outpatient units (Redit-2-Diag study) [3]. The rate of progression of DCKD shows considerable variability. Although it is generally characterized by a chronic progressive course related to several risk factors, in some subgroups of subjects with T2DM, DCKD progresses very rapidly [4]. Traditional modifiable risk factors include hyperglycemia and hypertension, but dyslipidemia may also have a significant contribution. Indeed, despite the achievement of recommended targets of blood glucose and blood pressure, a residual risk for DCKD remains due to dyslipidemia, of which hypertriglyceridemia is the most frequent abnormality [5,6,7]. Therefore, early detection and treatment are crucial because progressive DCKD is associated with unfavorable clinical outcomes, including end-stage kidney disease (ESKD), but also cardiovascular disease (CVD), and increased mortality [8]. Thus, the aim of this study was to examine dyslipidemia and other risk factors for DCKD in subjects with T2DM to provide support for prevention and treatment. Assessment of progression was based on retrospective routine glomerular filtration rate (GFR) measurements.

## 2. Materials and Methods

### 2.1. Subjects

The study was conducted from June 2015 to March 2016 and included consecutive adult T2DM subjects regularly attending hospital-based centers and clinics in Greece [3]. Within a period of 6 months prior to recruitment, laboratory data (urea, creatinine, albuminuria, albumin/creatinine ratio, HbA_1c_, lipids) were acquired for all participating patients. The study was conducted according to the guidelines of the Declaration of Helsinki and approved by the Ethics Committee of NIMTS Hospital (Protocol Number: 62/16-10-2013). All participants gave their informed consent. Assessment of progression was based on retrospective routine GFR measurements calculated according to MDRD (Modification of Diet in Renal Disease). CKD severity was staged according to KDIGO criteria (Kidney Disease: Improving Global Outcomes) [3]. All subjects were classified according to eGFR (mg/mL/1.73 m^2^) as follows: G1 > 90, G2 60–89, G3a 45–59, G3b 30–44, G4 15–29, and G5 < 15. They were also classified into the following categories according to the albumin/creatinine ratio (mg/g Cr): A1 < 30, A2 30–300, and A3 > 300. Categories G1A1 and G2A1 were considered normal. Given the size of our sample and the difficulty to adequately represent the 18 CKD stages, four new categories were coined as follows: normal kidney function, mild CKD, moderate CKD, and severe CKD (Table 1). More details of the materials and methods were described in a previous publication by our group [3].

### 2.2. Statistical Analysis

The statistical package R (version 3.2.5—R Core Team, Vienna, Austria) was used. For numerical variables, data are presented as means and standard deviations (SD). Stages for kidney function were regrouped into four (normal kidney function, mild, moderate, severe DCKD) and two (normal kidney function vs. DCKD) categories for modeling purposes. Univariate analysis was originally performed. Multiple analysis was employed to associate CKD risk with hemoglobin A_1c_ (HbA_1c_), triglycerides (TG), and blood pressure (BP) values. The proportional odds methodology was used to quantify the relationship of DCKD with these risk factors. Significant covariates were selected using backward stepwise selection. Significance was determined at 5% level (two-tailed *p* < 0.05).

## 3. Results

Participants were from the Redit-2-Diag study that examined 1759 subjects with a mean age of 68 years and a mean T2DM duration of 13 years [3]. Overall, 1372 subjects (77.9%) had dyslipidemia and 1363 (77%) had hypertension. Patients with DCKD compared with patients without DCKD had higher mean HbA_1c_ (7.2 vs. 7.0%), systolic blood pressure (SBP) (137.7 vs. 130.7 mm Hg) and TG (160.3 vs. 136.1 mg/dL), *p* < 0.01 in all. Regarding the other lipid parameters, the DCKD group had total cholesterol of 174.03 mg/dL, HDL-C 46.5 mg/dL, and LDL-C 95.9 mg/dL while the non-DCKD had 174.2 mg/dL, 47.7 mg/dL and 98.4 mg/dL levels, respectively, but with no statistical significance (Figure 1). An increase of HbA_1c_ (1 unit) and SBP (1 mm Hg) increased the probability of being classified into a higher CKD stage by 14% and 26% respectively. Moreover, an increase of TG by 88.5 mg/dL increased the risk by 24% (Figure 2). Regarding the antidiabetic treatment, most participants were on metformin (*n* = 1438, 81.8%), DPP-4 inhibitors (*n* = 716, 40.7%) and insulin (*n* = 716, 40.8%). Angiotensin-converting enzyme inhibitor (ACE-I) or an angiotensin II receptor blocker (ARB) were the most commonly prescribed antihypertensive medications, followed by calcium channel blockers (CCBs), B-blockers, and diuretics. For the management of dyslipidemia, most participants were treated with statins (*n* = 1690, 96%), followed by ezetimibe (*n* = 181, 10.2%). Details for the administration of antidiabetic, antihypertensive, and lipid-lowering medication were described in a previous publication by our group [9].

## 4. Discussion

Hyperglycemia and hypertension are traditional risk factors for DCKD in subjects with diabetes. These risk factors are also important for the development of CVD, which shares insulin resistance with DCKD as a common pathogenic mechanism [10,11]. These factors induce endothelial damage, leading to atherosclerosis in large and smaller kidney vessels [12,13]. Some of these effects also seem to contribute to the association of DCKD with abdominal aortic aneurysms [14]. Their contribution to the pathophysiology and progression of vascular disease is particularly important in earlier DCKD stages [15]. In our study, progression was assessed on the basis of retrospective routine GFR measurements.

Patients with T2DM and DCKD cluster the highest CVD risk. In a study involving over 1.2 million participants with or without T2DM, the presence of CKD was associated with an increased incidence of myocardial infarction, suggesting that DCKD should be added to the list of criteria defining people at the highest risk for coronary events [16]. Recently, a multicenter randomized controlled trial enrolling 395 patients with CDKD in primary cardiovascular prevention (NID-2 study) showed that a multifactorial intensive treatment of all major risk factors (hyperglycemia, dyslipidemia, hypertension) can effectively reduce MACEs and mortality within less than 3 years of intervention; interestingly, these CVD benefits persisted for over 13 years of follow-up [17].

In our study, an increase of HbA_1c_ by 1 unit increased the probability of being classified into a higher DCKD stage by 14%. The presence of DCKD affects glycemic control, making the therapeutic management of T2DM particularly challenging. Moreover, progressive loss of renal function impairs renal gluconeogenesis, whereas, insulin resistance can be triggered or worsened by elevated counter-regulatory hormones, altered lipid metabolism, electrolyte abnormalities, uremic acidosis, and accumulation of uremic toxins [18]. On the other hand, hyperglycemia is known to exert numerous deleterious effects on the kidney, including glomerular hyperfiltration, vasoconstriction, and increased advanced glycation end products (AGEs) [19,20,21]. The goals and treatment for the management of diabetes in DCKD are described in the 2020 KDIGO guidelines [22]. Hyperglycemia in the setting of obesity may also impair the process of autophagy, which can lead to renal tubular cell damage [23].

Moreover, an increase in SBP by 1 mm Hg in our patients increased the probability of being classified into a higher DCKD stage by 26% (Figure 2). In patients with DCKD, the optimal BP for minimizing the risk of DCKD progression and cardiovascular events is unclear [24]. Several guidelines recommend a goal of BP < 130/80 mm Hg and a BP < 140/90 mm Hg for most patients with T2DM and DCKD [25,26]. In a high percentage of subjects with T2DM, arterial hypertension is already present at the time of diagnosis [27]. In patients with DCKD stages 3 to 5, each 10 mm Hg increase in mean SBP has been associated with a 15% increase in the hazard ratio for the development of both micro- and macroalbuminuria and impaired kidney function [28]. In a recent study, the mean improvement in SBP of 11,7 mm Hg amongst a cohort of 143 patients with normal function or only mild DCKD significantly improved the serum creatinine over an average of 30 months, suggesting that the improvement could be due, at least in part, to the reduction in BP [29].

Subjects with diabetes and DCKD exhibit a characteristic lipid pattern of significant hypertriglyceridemia and low plasma levels of HDL-C [30]. In our study, 77.9% of the subjects had dyslipidemia. Our patients with DCKD compared with patients without DCKD had significantly higher plasma TG levels (160.3 vs. 136.1 mg/dL, *p* < 0.01); differences in plasma HDL-C (46.5 mg/dL vs. 47.7 mg/dL) and LDL-C levels (95.9 mg/dL vs. 98.4 mg/dL) were not statistically significant. An increase of TG by 88.5 mg/dL increased the risk of classification to a worse DCKD stage by 24%.

Several studies have recognized dyslipidemia as a risk factor in the development and progression of DCKD in T2DM, although its role is still unclear [31,32,33,34]. Over the last years, renewed attention to this topic has emerged. In a cohort of almost 2 million male US veterans followed for over nine years, examined the association between HDL-C and risks of incident CKD and CKD progression. The results showed the following: (a) an independent and gradual association between plasma HDL-C levels and eGFR deterioration, and (b) a U-shaped relationship between HDL-C levels and kidney disease progression as follows: low HDL-C (below 30 mg/dL) but also high HDL-C (above 60–70 mg/dL) were significantly associated with eGFR deterioration [35]. Similar results have been reported in patients with CKD followed for about three years, as follows: either low or high plasma levels of total cholesterol and high LDL-C levels were significant risk factors for rapid development and progression of CKD [36]. In the AMD Annals Initiative Study, Russo et al. investigated the contribution of high TG and/or low HDL-C to the residual risk for DCKD in 15.362 patients with T2DM attending diabetes centers throughout Italy over a period of four years; most of the patients were on anti-diabetic (oral agents and/or insulin), lipid-lowering (statins, fibrates), and anti-hypertensive medications, and aspirin. The results of this elegant study supported the independent role of TG/HDL-C in the development and progression of DCKD and showed that, although LDL-C levels were well controlled, TG above 150 mg/dL increased the risk of kidney failure/eGFR reduction by 35%, and HDL-C below 40 mg/dL in men and 50 mg/dL in women increased this risk by 44%; low HDL-C and high TG levels in these patients were associated with higher BMI and HbA_1c_. The differences in TG/HDL-C levels between the DCKD and no-DCKD cohorts were attenuated but remained significant after correcting for other major risk factors such as HbA_1c_, BP, and plasma LDL-C levels [37]. Another study on newly-onset T2DM provided convincing evidence that the long-term TG control was particularly important in delaying the decline of kidney function in the early stages of diabetic kidney disease [38]. In addition to the potential role of dyslipidemia and diabetes in the progression of DCKD, diabetic nephropathy per se can also impair lipid metabolism, mainly via urinary protein excretion [39]. However, we should point out that, to the best of our knowledge, there are no prospective RCTs demonstrating that lowering blood lipid levels results in slower DCKD progression; all available data comes from retrospective observational analyses.

The role of insulin resistance in the development and progression of DCKD deserves some discussion. Insulin resistance is a major pathophysiological mechanism in the development of hyperglycemia, hyperinsulinemia, and dyslipidemia in patients with T2DM [40]. Moreover, the kidney plays an important role in glucose homeostasis, and increased rates of renal gluconeogenesis make a significant contribution to hyperglycemia in T2DM, particularly under conditions of acidosis [41]. Insulin resistance develops in parallel to the impairment of renal function, while diminished renal insulin clearance may contribute to hyperinsulinemia; the kidney is responsible for 10–20% of insulin clearance from the systemic circulation [42,43]. There is evidence supporting a pathogenetic role of insulin resistance in the progressive deterioration of kidney function via glomerular hyperfiltration and increased vascular permeability caused by hyperinsulinemia, subclinical inflammation, and podocyte abnormalities [44,45]. Clinical evidence suggests that progressive kidney dysfunction changes the composition and quality of blood lipids, mainly HDL-C and TG-rich lipoproteins, in favor of a more atherogenic profile. Several factors modify the composition of HDL-C particles in DCKD, including insulin resistance, increased oxidative stress, a proinflammatory microenvironment, and uremic toxins [46]. Hypertriglyceridemia may result from the following several defects: (a) increased availability of non-esterified fatty acids to the liver due to insulin resistance and increased rates of lipolysis in the adipose tissue [41,47,48]. (b) Decreased degradation of VLDL triglycerides due to the hypoactive hepatic triglyceride lipase. An increased number of lipase inhibitors (apo C-III) in the setting of uremia may also play a role in the decrease in lipoprotein lipase-dependent TG-rich lipoprotein catabolism [49]. (c) In subjects with T2DM, obesity, and insulin resistance, hypertriglyceridemia has been shown to be attributed to a defect in the postprandial dynamic adjustment of triglyceride clearance across the adipose tissue, caused by blunted insulin-stimulated rates of blood flow [48,50,51]. TG-rich lipoproteins are important risk factors for cardiovascular outcomes in DCKD patients [52]. Regarding the role of LDL-C in DCKD, despite relatively lower levels, the oxidation of LDL-C due to inflammation and increased oxidative stress may account for its greater atherogenic effect [53].

In our study, T2DM subjects were on medications to treat hyperglycemia (most commonly metformin, DPP-4 inhibitors, and basal insulin), hypertension (most commonly angiotensin-converting enzyme inhibitors or angiotensin II receptor blockers), and dyslipidemia (most commonly statins). Anti-hyperglycemic medications reduce glucotoxicity and lipotoxicity (decrease plasma non-esterified fatty acid levels) and, therefore, ameliorate insulin resistance [54]. Since hypertriglyceridemia in T2DM is due, at least in part, to insulin resistance [48,50,51], the use of anti-hyperglycemic drugs could interfere to some extent with the clinical outcomes in the present study. However, this would not affect the conclusions since plasma triglyceride levels were significantly increased in the group of T2DM with DCKD versus the group without DCKD (Figure 1), and hypertriglyceridemia was involved in the progression of kidney failure as estimated by GFR (Figure 2). The use of statins for the treatment of dyslipidemia in our study may explain the findings that HDL-C levels were not statistically different between the groups with or without DCKD. Several reports have suggested that statins may affect insulin sensitivity, but the results are controversial, showing either a beneficial, a neutral, or a worsening effect depending on the type/dose of these drugs and the co-existence of obesity [55,56]. In our study, plasma HDL-C levels in the two groups were ~40–50 mg/dL; at these levels, HDL-C has been shown not to affect the risk of renal outcomes [35]. Finally, anti-hypertensive medications used in subjects with T2DM and DCKD may also ameliorate insulin resistance and, therefore, delay the progression of kidney failure [57,58].

The strengths of this study are the nationwide/real-world design; our patients were regularly screened in diabetes centers all over the country, and, therefore, our study is representative of common clinical practice. The limitations include the lack of centralized measurements and the cross-sectional nature of our study.

## 5. Conclusions

Hyperglycemia and hypertension are strong traditional risk factors for the development and progression of DCKD. Despite the achievement of recommended targets for blood glucose and BP, the residual risk of DCKD remains relatively high. Hypertriglyceridemia is also common in subjects with T2DM. In our study, increases in HbA_1c_ by 1%, SBP by 1 mm Hg, and plasma TG levels by 88.5 mg/dL increased the progression of DCKD by 14%, 26%, and 24%, respectively. Therefore, in addition to hyperglycemia (a marker of glycemic/diabetes control) and hypertension, elevated TG levels should be aggressively treated to prevent DCKD progression in T2DM. However, it should be pointed out that our findings refer to associations between calculated GFR and measurements of plasma glucose/lipid levels, BP, and HbA_1c_. Therefore, long-term prospective studies are required to confirm our conclusions.

## Figures and Tables

**Figure 1 jcm-11-03224-f001:**
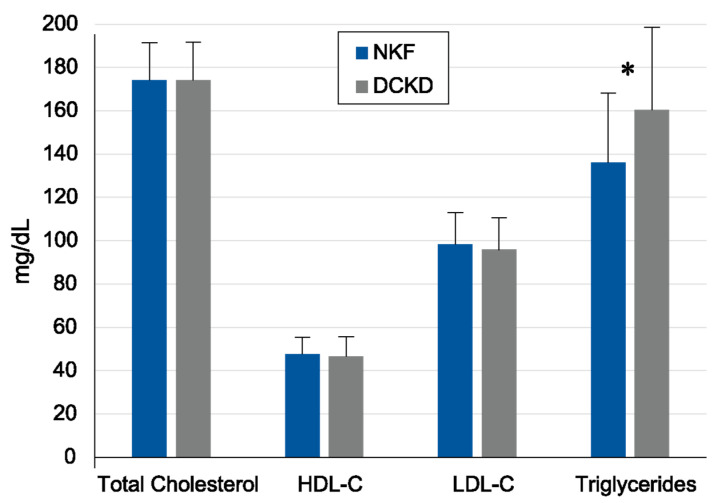
Plasma concentrations of lipid levels in subjects with T2DM and normal kidney function (NKF) or diabetic chronic kidney disease (DCKD). * *p* < 0.01.

**Figure 2 jcm-11-03224-f002:**
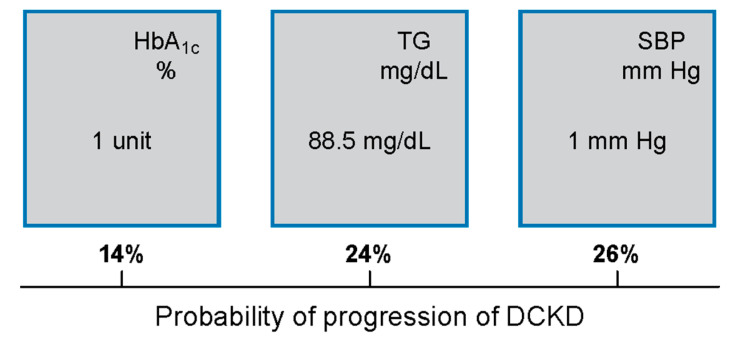
An increase of HbA_1c_ by 1 unit, triglycerides (TG) by 88.5 mg/dL, and systolic blood pressure (SBP) by 1 mm Hg, increased the progression of diabetic chronic kidney disease (DCKD) by 14%, 24%, and 26%, respectively.

**Table 1 jcm-11-03224-t001:** Disease characteristics. Two-group analysis (normal kidney function vs. DCKD). Values are presented as a mean ± standard deviation (SD). DCKD: Diabetic Chronic Kidney Disease, SBP: Systolic Blood Pressure, DBP: Diastolic Blood Pressure, HDL-C: High Density Lipoprotein-Cholesterol, LDL-C: Low Density Lipoprotein-Cholesterol, TG: Triglycerides).

Redit-2-DiagSample, *n*	Normal Kidney Function965	Mild DCKD539	Moderate DCKD135	Severe DCKD120	Total1759	Odds	95% CI
Mean HbA_1c_, %, (SD)	7.0 (1.1)	7.1 (1.2)	7.2 (1.2)	7.3 (1.3)	7.1 (1.2)	1.14	1.06–1.23
Mean SBP, mm Hg, (SD)	130.7 (14.9)	136.1 (16.7)	138.7 (17.9)	138.4 (18.7)	135.9 (17.0)	1.26	1.19–1.33
Mean DBP, mm Hg, (SD)	76.9 (9.2)	77.4 (10.6)	77.4 (10.9)	75.9 (10.4)	76.9 (9.9)	1.01	0.92–1.11
Total cholesterol, mg/dL, (SD)	174.2 (38.0)	176.4 (39.3)	171.3 (33.8)	174.4 (43.4)	174.0 (38.6)	1.00	1.00–1.00
HDL-C, mg/dL, (SD)	47.7 (17.0)	47.1 (12.9)	44.8 (11.6)	47.8 (34.9)	46.8 (19.1)	1.00	0.99–1.00
LDL-C, mg/dL, (SD)	98.4 (32.0)	98.8 (32.9)	94.6 (29.6)	94.3 (35.0)	96.5 (32.8)	1.00	1.00–1.00
TG, mg/dL, (SD)	136.1 (69.9)	143.4 (71.6)	162.8 (78.2)	174.8 (100.7)	154.2 (80.1)	1.36	1.22–1.52

## Data Availability

The data presented in this study are available upon request from the corresponding author.

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
