# Peer review of "Hypertriglyceridemia and Other Risk Factors of Chronic Kidney Disease in Type 2 Diabetes: A Hospital-Based Clinic Population in Greece"

_jcm, 2022, doi:10.3390/jcm11113224_

Round 1
Reviewer 1 Report
The manuscript describes the study regarding risk factors of diabetic chronic kidney disease. Despite the problem is very common and widely studied presented results are still interesting while the study included the large number of patients from different centers.
I have only one comment - in Abstract - first line "prevalence of chronic disease (CKD)" word kidney is missing.
Author Response
The manuscript describes the study regarding risk factors of diabetic chronic kidney disease. Despite the problem is very common and widely studied presented results are still interesting while the study included the large number of patients from different centers.
I have only one comment - in Abstract - first line "prevalence of chronic disease (CKD)" word kidney is missing.
Thank you for your comment, error has been corrected.
Reviewer 2 Report
Comments to author
Manuscript ID: jcm-1688506
Dear authors,
This is an interesting manuscript and the data presented are relevant to the field of study. However, there are some issues to be addressed. Some suggestions with helpful feedback on the manuscript are below.
Major comments/concerns:
Materials and Methods
The authors state that “the study was approved by the institutional committees and participants gave their informed consent”. However, an institutional review board approval (number) is required, and it must be informed in the Materials and Methods section, regardless of whether such details have been published in a previous independent study.
Materials and Methods section is very brief, a more detailed description must be informed in this section, regarding sample, data collection and conditions/variables, regardless of whether such details have been published in a previous independent study.
Statistical analysis must be better clarified, in which each informed test associates with each p value informed in results section.
Results
Why are the results for HBA1c, systolic blood pressure and TG not shown in a figure or table with the respective standard deviation (SD) or standard error (SE)? It would be interesting to present them in a figure or table. This is further justified by the fact that these are the most important results that were statistically significant (p < 0.05).
Figure 1: Standard deviation (SD) or standard error (SE) must be informed.
Despite being vaguely informed in the methodology, which only leaves one to understand (and which reinforces the need for more details in the Materials and Methods section), in figure 1 the term NKF can confuse the reader and it should be clarified if it refers to patients with DM who did not develop CKD (which seems more likely due to the brief description of the sample in the methodology) or if it refers to patients without DM and without CKD (which is implied in the figure, but not in the methodology). This should be clarified to facilitate the reader's understanding.
Figure 2: the adopted classification (stratification) of CKD stage must be better clarified/described in the Materials and Methods section and after discussed in the discussion section.
Discussion
A better description of the sample could contribute a lot to the study, as it could even better foster discussion. For example, patients under medication to treat dyslipidemia could interfere with the clinical outcome of the groups evaluated in the present study, as well as medication to control T2DM or systolic blood pressure.
Despite being mentioned in the “introduction section” as a possible item to be included in the risk stratification for DCKD, the lipid profile in most parameters did not present significant results in the present study (total cholesterol, LDL and HDL). The authors could discuss this further in the Discussion section, highlighting why only triglycerides (and not also high levels of total cholesterol, LDL and/or low HDL levels) showed significant results.
Discuss better the results for HBA1c (marker of glycemic/DM control) and systolic blood pressure.
Conclusion
In the sentence: “Hyperglycemia and hypertension are strong risk factors for CKD and CVD”. Remove CVD (cardiovascular disease), as no direct parameters associated with direct outcomes in the cardiovascular system were evaluated in this study. Only “variables” (including some that may have serious consequences for CVD) associated with CKD progression were evaluated.
The same can be seen for the sentence: “Dyslipidemia and predominant hypertriglyceridemia, which is common in subjects with T2DM and correlated significantly with DCKD and CVD”. Remove the general term “Dyslipidemia”, as only hypertriglyceridemia was found in the present study, and there was no significance for total cholesterol, LDL and HDL.
Highlight the variables HBA1c (marker of glycemic/DM control) and systolic blood pressure, which proved to be significant.
Minor comments/concerns:
Abstract: ...increasing prevalence of “chronic disease (CKD)”..... Please, something is missing in this sentence. Did the authors mean “chronic kidney disease (CKD)”?
English should be improved.
Author Response
This is an interesting manuscript and the data presented are relevant to the field of study. However, there are some issues to be addressed. Some suggestions with helpful feedback on the manuscript are below.
Major comments/concerns:
Materials and Methods
- The authors state that “the study was approved by the institutional committees and participants gave their informed consent”. However, an institutional review board approval (number) is required, and it must be informed in the Materials and Methods section, regardless of whether such details have been published in a previous independent study.
The institutional review board approval number is now provided.
- Materials and Methods section is very brief, a more detailed description must be informed in this section, regarding sample, data collection and conditions/variables, regardless of whether such details have been published in a previous independent study.
The “Materials & Methods” section has been expanded according to the reviewer’s suggestion
- Statistical analysis must be better clarified, in which each informed test associates with each p value informed in results section.
Statistical analysis has been better clarified according to the reviewer’s suggestion
Results
- Why are the results for HBA1c, systolic blood pressure and TG not shown in a figure or table with the respective standard deviation (SD) or standard error (SE)? It would be interesting to present them in a figure or table. This is further justified by the fact that these are the most important results that were statistically significant (p < 0.05).
A Table with these results is now provided
- Figure 1: Standard deviation (SD) or standard error (SE) must be informed.
Standard deviations are now provided in Figure 1.
- Despite being vaguely informed in the methodology, which only leaves one to understand (and which reinforces the need for more details in the Materials and Methods section), in figure 1 the term NKF can confuse the reader and it should be clarified if it refers to patients with DM who did not develop CKD (which seems more likely due to the brief description of the sample in the methodology) or if it refers to patients without DM and without CKD (which is implied in the figure, but not in the methodology). This should be clarified to facilitate the reader's understanding.
The reviewer is correct, NKF refers to patients with DM and normal kidney function. This is now clarified in Figure 1.
- Figure 2: the adopted classification (stratification) of CKD stage must be better clarified/described in the Materials and Methods section and after discussed in the discussion section.
The stratification of CKD is now better clarified in the “Materials and Methods” section as requested by the reviewer. We also changed the legend of Figure 2 as follows: “An increase of HbA1c by 1 unit, triglycerides (TG) by 88.5 mg/dL, and systolic blood pressure (SBP) by 1 mmHg, increased the progression of chronic kidney disease (CKD) by 14%, 24% and 26%, respectively”.
Discussion
- A better description of the sample could contribute a lot to the study, as it could even better foster discussion. For example, patients under medication to treat dyslipidemia could interfere with the clinical outcome of the groups evaluated in the present study, as well as medication to control T2DM or systolic blood pressure.
Following the reviewer’s suggestion, we now provide the information regarding the treatment of our patients in the “Materials and Methods”. We have also added a paragraph to discuss the possibility that treatment for hyperglycemia, dyslipidemia, mainly hypertriglyceridemia and hypertension could affect the clinical outcomes, as requested by the reviewer.
- Despite being mentioned in the “introduction section” as a possible item to be included in the risk stratification for DCKD, the lipid profile in most parameters did not present significant results in the present study (total cholesterol, LDL and HDL). The authors could discuss this further in the Discussion section, highlighting why only triglycerides (and not also high levels of total cholesterol, LDL and/or low HDL levels) showed significant results.
As requested by the reviewer, we have added a paragraph in the discussion to address this point.
- Discuss better the results for HBA1c (marker of glycemic/DM control) and systolic blood pressure.
This has been done.
Conclusion
- In the sentence: “Hyperglycemia and hypertension are strong risk factors for CKD and CVD”. Remove CVD (cardiovascular disease), as no direct parameters associated with direct outcomes in the cardiovascular system were evaluated in this study. Only “variables” (including some that may have serious consequences for CVD) associated with CKD progression were evaluated.
CVD has been removed
- The same can be seen for the sentence: “Dyslipidemia and predominant hypertriglyceridemia, which is common in subjects with T2DM and correlated significantly with DCKD and CVD”. Remove the general term “Dyslipidemia”, as only hypertriglyceridemia was found in the present study, and there was no significance for total cholesterol, LDL and HDL.
Dyslipidemia has been removed.
- Highlight the variables HBA1c (marker of glycemic/DM control) and systolic blood pressure, which proved to be significant.
This has been done.
Minor comments/concerns:
Abstract: ...increasing prevalence of “chronic disease (CKD)”..... Please, something is missing in this sentence. Did the authors mean “chronic kidney disease (CKD)”?
Grammatical error has been corrected.
English should be improved.
English has been improved as requested by the reviewer.
Reviewer 3 Report
This manuscript presents several issues that should be addressed.
1- In this short article, some chapters are too succinct. In particular, materials and methods, the results and limitations of the study must be enriched and implemented.
2- This cross-sectional study conducted in 2015/16 is submitted for publication in 2022. Authors should comment on this about six-year interval.
3- In Conclusion the authors state "Hyperglycemia and hypertension are strong risk factors for CKD and CVD. Despite the better control of these traditional risk factors, the residual risk for DCKD is still high." Actually diabetic patients with CKD cluster the highest CV risk. A big challenge for National Health Systems is to reduce this risk. Very recently, the NID-2 study has shown in a multicentre randomized controlled trial that a in diabetic nephropathic population in primary cardiovascular prevention, a multifactorial intervention can reduce MACEs and mortality in a few years and with long durability (Cardiovasc Diabetol (2021) 20:145. doi: 10.1186/s12933-021-01343-1). This hot issue should be added in discussion.
4- Figure 1 describes the group of patients with normal renal function (NKF). In the manuscript, reference is never made to the NKF. What do the authors mean by NFK? A GFR> 60ml/min/1.73m2? Please clarify.
5- A linguistic revision by a native English speaker is required.
Author Response
This manuscript presents several issues that should be addressed.
- In this short article, some chapters are too succinct. In particular, materials and methods, the results and limitations of the study must be enriched and implemented.
The relevant sections have been enriched as requested by the reviewer.
- This cross-sectional study conducted in 2015/16 is submitted for publication in 2022. Authors should comment on this about six-year interval.
It should be appreciated that after the collection of the data it took us a considerable amount of time to make the statistical analysis, organize the three publications out of this work and most importantly to arrange the logistics and find the financial resources to cover the expenses. The latter was not easy at all. This work was not supported by University Grants but by the Hellenic Diabetes Association and all the bureaucracy involved. On top of this we had the pandemic over the last two years that disorganized the whole country, as in the rest of the world. We hope the reviewer can understand the reasons for the delay.
- In Conclusion the authors state "Hyperglycemia and hypertension are strong risk factors for CKD and CVD. Despite the better control of these traditional risk factors, the residual risk for DCKD is still high." Actually diabetic patients with CKD cluster the highest CV risk. A big challenge for National Health Systems is to reduce this risk. Very recently, the NID-2 study has shown in a multicentre randomized controlled trial that a in diabetic nephropathic population in primary cardiovascular prevention, a multifactorial intervention can reduce MACEs and mortality in a few years and with long durability (Cardiovasc Diabetol (2021) 20:145. doi: 10.1186/s12933-021-01343-1). This hot issue should be added in discussion.
The results of this important study have been included in the discussion.
- Figure 1 describes the group of patients with normal renal function (NKF). In the manuscript, reference is never made to the NKF. What do the authors mean by NFK? A GFR> 60ml/min/1.73m2? Please clarify.
The reviewer is correct, NKF describes patients with T2DM without nephropathy, this is now defined in the legend of the Figure
- A linguistic revision by a native English speaker is required.
This has been done as requested by the reviewer.
Round 2
Reviewer 3 Report
No further comments.
Author Response
- This is a useful paper but in parts is a bit confused about “risk factors”. I think they are examining risk factors usually related to cardiovascular disease as factors for CKD progression. In their introduction and their discussion, they need to make it clear which disease process they are referring to at all times.
ANSWER: We have clarified this issue in the beginning of the discussion, as follows: “Hyperglycemia and hypertension are traditional risk factors for DCKD in subjects with diabetes; these risk factors are also important in the development of CVD which shares insulin resistance with DCKD as a common pathogenic mechanism.” (page 4; marked in red)
- The paper refers to CKD progression without defining or quantifying progression. It needs to be made clear that the assessment of progression was based on retrospective routine GFR measurements.
ANSWER: We thank the Editor for this comment. Throughout the Introduction, Methods, and Discussion we have now rendered it clear that progression was assessed on the basis of retrospective routine GFR measurements (marked in red).
- Clear criteria for progression are needed in the methods and/or stats sections. In their review of previous work looking at this area, it is also worth clarifying that there are no prospective RCTs that demonstrate that lipid lowering results in slower CKD progression, all available data are retrospective observational analyses (to the best of my knowledge).
ANSWER: Again, we thank the Editor for this comment. We agree with the Editor and have added his/her comment in the discussion (page 5; marked in red).
- The section on BP control is better in that it acknowledges the lack of clear evidence on BP control and progression and the lack of a clear BP target.
ANSWER: We thank the Editor for this comment.
- Finally, the conclusion is poorly worded. These are associations only. Are they really referring to intra-subject “increases” in values or in higher or lower mean values? Some careful re-writing would very much help the reader to take away the correct messages.
ANSWER: We have added the following sentence: “However, it should be pointed out that our findings refer to associations between estimated GFR and measurements of plasma glucose/lipid levels, BP and HbA1c; therefore, long-term prospective studies are required to confirm our conclusions (page 6; marked in red).
- In the text sent to us, we see a red line across Figure 2; it is not clear what this means. Does the Editor wish to make changes in this Figure?